# Neoadjuvant Immunotherapy: A Promising New Standard of Care

**DOI:** 10.3390/ijms241411849

**Published:** 2023-07-24

**Authors:** Emma Boydell, Jose L. Sandoval, Olivier Michielin, Michel Obeid, Alfredo Addeo, Alex Friedlaender

**Affiliations:** 1University Hospital of Geneva, 1205 Geneva, Switzerland; 2University Hospital of Lausanne, 1005 Lausanne, Switzerland; 3Clinique Générale Beaulieu, 1206 Geneva, Switzerland

**Keywords:** immunotherapy, neoadjuvant, predictive biomarkers, NSCLC, melanoma, urothelial carcinoma, gastric cancer

## Abstract

Neoadjuvant immunotherapy has emerged as a promising approach in the treatment of various malignancies, with preclinical studies showing improved immune responses in the preoperative setting. FDA-approved neoadjuvant-immunotherapy-based approaches include triple-negative breast cancer and early non-small cell lung cancer on the basis of improvement in pathological response and event free survival. Nevertheless, current trials have only shown benefits in a fraction of patients. It is therefore crucial to identify predictive biomarkers to improve patient selection for such approaches. This review aims to provide an overview of potential biomarkers of neoadjuvant immunotherapy in early triple-negative breast cancer, bladder cancer, melanoma, non-small cell lung cancer, colorectal cancer and gastric cancer. By the extrapolation of the metastatic setting, we explore known predictive biomarkers, i.e., PD-L1, mismatch repair deficiency and tumour mutational burden, as well as potential early-disease-specific biomarkers. We also discuss the challenges of identifying reliable biomarkers and the need for standardized protocols and guidelines for their validation and clinical implementation.

## 1. Introduction

Immune checkpoint inhibitors (ICIs) targeting programmed death 1 (PD-1) and its ligand, programmed death ligand 1 (PD-L1), as well as cytotoxic T-lymphocyte antigen 4 (CTLA-4) have become the standard of care for many tumour types and are now indicated in approximately half of US patients with advanced or metastatic cancer [1,2]. ICIs were initially tested in the metastatic setting, and in light of high response rates and prolonged survival, ICI trials were expanded to earlier settings, in which patients often face high relapse rates. In the adjuvant setting, ipilimumab was the first ICI to be approved by the FDA for resected stage III melanoma [3]. Neoadjuvant management has the potential to reduce tumour volume preoperatively for short-course treatments and therefore limit postoperative relapse and surgery-related morbidity.

Preclinical studies have shown that preoperative immunotherapy leads to increased survival compared to adjuvant immunotherapy. For instance, to test this hypothesis, Liu et al. used a rapidly metastasizing triple-negative breast cancer (TNBC) model in mice in which regulatory T cells were conditionally depleted, mimicking immunotherapy. Survival was significantly increased in the mice depleted of regulatory T cells (T-reg) prior to surgery than in those with T-reg depletion after surgery. Neoadjuvant treatment with antiPD-1 and antiPD-L1 in these mice also significantly increased survival, yet fewer long-term survivors were seen than in the T-reg-depleted mice, suggesting that a combination treatment may be required to obtain a longer response. Furthermore, the long-term survivors presented no residual latent tumour cells, and the effect of neoadjuvant ICIs relied on an intact IFN-gamma pathway and, to a lesser extent, on CD8, CD4 and NK cells.

One of the hypotheses for the difference in efficacy between neoadjuvant and adjuvant immunotherapy is an increase in tumour-specific effector or memory T cells after neoadjuvant treatment. It has also been suggested that necrosis of the primary tumour may lead to an increased shedding of tumour-specific antigens and, therefore, have the potential to induce systemic priming and the expansion of antigen-presenting cells [4]. This mechanism could ultimately enhance the clearance of micrometastatic disease, the main culprit of relapse after surgery. Immune checkpoint inhibitors (ICIs) are also less toxic than chemotherapy and can provide immunologic memory, resulting in durable clinical benefits [5]. In addition, neoadjuvant treatment may prove beneficial because early-stage lung cancers may have a higher clonal neoantigen burden, which has been shown to correlate with improved survival compared to that of patients with more heterogeneous (subclonal) neoantigens within their tumours [6,7]. Subsequent work from Liu et al. showed an ideal timing for neoadjuvant therapy, and the benefit is diminished if the treatment is shortened or prolonged [8]. Furthermore, the addition of adjuvant immunotherapy after neoadjuvant immunotherapy did not increase survival.

These data have supported the development of many clinical trials, including the phase III KEYNOTE-522 and CHECKMATE-816 trials, which led to the FDA approval of neoadjuvant pembrolizumab in early triple-negative breast cancer and neoadjuvant nivolumab in resectable non-small cell lung cancer (NSCLC). Both studies reported an increase in pathologic complete response rates (pCR, no visible residual tumour). Other studies have also reported major pathological response rates (MPR, <10% residual viable tumour).

Identifying patients who will benefit from such an approach depends on reliable biomarkers. Two types of biomarkers are used in oncology: prognostic and predictive biomarkers. Prognostic biomarkers are associated with a favourable outcome regardless of treatment in the biomarker-positive versus -negative population due to a different intrinsic process and the spontaneous natural evolution of the tumour. Some have been proposed as possible stratification factors for future trial design [9,10,11,12]. Predictive biomarkers, on the other hand, confer a greater treatment effect in the biomarker-positive population than in the biomarker-negative population, implying that both populations will have the same outcome without treatment [13]. To date, in the metastatic setting, PD-L1 and mismatch repair deficiency (dMMR)/microsatellite instability (MSI-H) are the only known positive predictive biomarkers [14,15]. Other biomarkers, such as the tumour mutational burden (TMB), interferon-gamma signatures and tumour-infiltrating lymphocytes (TILS), among others, are biologically plausible but have yet to show a conclusive correlation with clinical outcomes. In the case of the TMB for example, the biological rationale is similar to that of dMMR (with high rates of mutations leading to more neoantigens, making the tumour more immunogenic). The FDA has approved pembrolizumab in advanced tumours harbouring a TMB > 10 mut/Mb based on increased response rates [15]. However, this approval has been controversial for several reasons, namely the use of a seemingly arbitrary cutoff of 10mut/Mb and the reliance on an endpoint that is not clinically meaningful [16].

This review provides an overview of potential biomarkers for neoadjuvant immunotherapy across numerous cancer types, including NSCLC, triple-negative breast cancer, melanoma, bladder cancer, gastric cancer and colorectal cancer. By extrapolation of the metastatic setting, we will explore known biomarkers and potential early-disease-specific biomarkers.

## 2. Non-Small Cell Lung Cancer

Approximately half of patients diagnosed with early resectable NSCLC experience tumour recurrence after surgery [17]. Adjuvant chemotherapy is the standard of care after resection in patients with stage II and III disease and provides an 8.6% survival benefit at 5 years compared to surgery alone [18,19]. Chemotherapy in the neoadjuvant setting has been tested in several trials and provides similar survival benefits as in the adjuvant setting but is not the standard of care [20]. pCR rates with neoadjuvant chemotherapy have been reported at 6 to 8% [21]. To further improve outcomes, several trials have been conducted to evaluate the role of neoadjuvant ICIs, alone or in combination with chemotherapy.

Single-agent ICIs in NSCLC in the neoadjuvant setting were first tested in the CHECKMATE 159 pilot trial, which reported an MPR of 45% in patients with stage I–IIIA NSCLC treated with two preoperative doses of nivolumab [22].

Neoadjuvant treatment-related adverse events (TRAEs) were observed in five patients, but only one experienced pneumonia at a grade ≥ 3. There were no treatment-related surgical delays, and the median interval between the administration of the second dose of nivolumab and surgery was 18 days (with an interquartile range (IQR) of 11–29 days).

Subsequently, Gao et al. found similar results for resectable NSCLC patients who received two cycles of neoadjuvant sintilimab (a PD-1 inhibitor) [23]. In this study, 40 patients with stage IA–IIIB NSCLC received two doses of sintilimab, and 37 underwent radical resection. Of the thirty-seven patients, fifteen patients (40.5%, 95% CI: 24.8–57.9%) achieved an MPR, and six (16.2%, 95% CI: 6.2–32.0%) achieved a pathologic complete response (pCR). Immune-related adverse events (IRAEs) were observed in twenty-one patients (52.5%), of whom four (10%) experienced a grade ≥ 3 adverse event and one died.

In 2021, a phase II trial (LCMC3, NCT0292730) analysed the effect of two cycles of neoadjuvant atezolizumab (in 181 patients with stage IB–IIIB NSCLC) [24]. Of the 181 patients, 159 (88%) underwent surgical resection, and the median time from the end of the neoadjuvant therapy to surgery was 22 days. A total of 20.4% (95% CI: 14–28%) achieved an MPR, and 6.8% (95% CI: 3–12%) achieved a pCR. Immunotherapy-related AEs were observed in only 24.3% of the patients, of which only 2.2% were TRAEs with a grade ≥ 3. In the phase II PRINCEPS clinical trial, 30 patients with resectable clinical stage IA (≥2 cm)–IIIA, non-N2 NSCLC were treated with neoadjuvant atezolizumab [25]. Among the patients, four (14%) achieved an MPR, and twelve (41%) had <50% residual tumour cells. Interim results were presented from another study, IONESCO, which evaluated neoadjuvant durvalumab [26]. Forty-six eligible patients with stage IB–IIIA, non-N2 resectable NSCLC were enrolled in this study, of whom forty-three underwent surgery. The primary endpoint was the rate of complete surgical resection (R0), which was achieved in 90% (41/43) of the patients who underwent surgery. The pathologic analysis showed that 18.6% of the patients achieved an MPR, and 7% achieved a pCR.

To further explore the potential of neoadjuvant immunotherapy in treating NSCLC, studies have been conducted examining the effect of ICIs in combination. A pivotal study by Cascone et al. evaluated the effect of neoadjuvant nivolumab or nivolumab plus ipilimumab followed by surgery in 44 patients with operable NSCLC. Of the 44 patients, 41 completed the planned neoadjuvant therapy, and 37 underwent surgery [27]. Of the 37 patients who underwent resection, 24% (5/21) and 50% (8/16) achieved an MPR in the nivolumab and the nivolumab plus ipilimumab arm, respectively. The pCR rate in the nivolumab plus ipilimumab arm (38%) was higher than in the nivolumab arm (10%). There were no major safety differences between nivolumab and nivolumab plus ipilimumab, although the sample size was small.

Regarding the combination of chemotherapy and ICIs, the NADIM, an open-label single-arm, phase II trial evaluated the impact of neoadjuvant nivolumab with carboplatin and paclitaxel, followed by adjuvant intravenous nivolumab monotherapy for 1 year in stage III resectable NSCLC [28]. Forty-six patients with stage IIIA (N2 or T4N0) NSCLC were enrolled in the study. An MPR was achieved in 83% (95% CI: 68–93%) of the patients, of whom 63% (95% CI: 62–91%) achieved a pCR. The 2-year progression-free survival (the intent-to-treat (ITT) population), which was the primary endpoint of the study, was 77.1% (95% CI: 59.9–87.7%).

Another interesting single-arm open-label phase II study was presented by the SAKK lung group, which used standard neoadjuvant cisplatin and docetaxel in combination with neoadjuvant durvalumab in patients with stage IIIA (N2) NSCLC [29]. In total, of the 68 patients who were enrolled in the study, 55 (80.9%) underwent resection and 62% of these patients achieved an MPR, of whom 18.2% achieved a pCR. The 1-year event-free survival (EFS), which was the primary endpoint of the study, was 73.3% (90% CI: 60.1–82.7%), and 12.9% of the patients experienced grade 3–4 neoadjuvant durvalumab-related AEs.

The phase III CHECKMATE 816 trial investigated preoperative nivolumab and platinum-based chemoimmunotherapy versus chemotherapy alone and reported pCR rates of 24% versus 2.2% and 31.6 months versus 20.8 months for event-free survival, respectively [30]. These results led to the FDA approval of neoadjuvant nivolumab in combination with chemotherapy for adult patients with resectable NSCLC (tumours ≥ 4 cm or node-positive).

Despite extensive post hoc analyses of the aforementioned trials, no significant predictive biomarker of response has been identified. While PD-L1 expression is positively correlated with survival in the metastatic setting, such findings were not reproduced in early NSCLC [31]. PD-L1-positive patients in the CHECKMATE816 trial had an improved EFS rate compared to that of PD-L1-negative patients (HR 0.41 versus 0.85) [30]. PD-L1 was also correlated with response in the LCMC3 (phase II, neoadjuvant atezolizumab) trial and in the NEOSTAR trial (phase II, neoadjuvant ipilimumab plus nivolumab) [27,32]. In contrast, in the NADIM (phase II, nivolumab plus chemotherapy) and SAKK 16/14 trials (phase II, perioperative durvalumab with cisplatin docetaxel), there was no association between response and PD-L1 expression [29,33]. Moreover, in a phase II test of a neoadjuvant atezolizumab and chemotherapy regimen (NCT02716038), patients achieved an MPR regardless of PD-L1 expression [34].

Besides PD-L1, other putative genomic and phenotypic biomarkers have been studied. Regarding histology, non-squamous subtypes treated with nivolumab had a greater EFS rate increase than squamous subtypes (HR 0.50 vs. 0.77) in the CHECKMATE-816 trial [30]. These findings contradict the results from the LCMC3 and NCT02716038 trials (neoadjuvant atezolizumab plus chemotherapy) [32,34].

As for the TMB, it was not correlated to response in the NADIM, LCMC3 or CHECKMATE-159 trials [30,32,33]. In CHECKMATE-816, the response to neoadjuvant chemoimmunotherapy was observed regardless of the TMB levels; however, the MPR rates were increased in patients with a TMB > 12.3 mutations/mB [30].

In LCMC3, peripheral blood CD8 T cell expansion was identified in pre- and post-treatment analyses. A 60-marker flow cytometry panel was developed to analyse baseline immune cell characteristics and potential correlations with the MPR. Interestingly, the model identified an association between responses and some NK and NK-like T cell subtypes. Higher levels of NK and NK-like T cells expressing inhibitory receptors, such as immunoglobulin-like transcript 2 (ILT2), NK group 2 member A and NK group 2 member D (NKG2A and NKG2D), were negatively correlated to the MPR [32]. These receptors have been associated with NK cell inhibition and immune tolerance [35]. Single-cell RNA sequencing in tissue samples identified a higher expression of NKG2A and killer cell immunoglobulin-like receptor (KIR) in NK cells in non-responders. A significant decrease in peripheral ILT2+ NKG2A+ KIR2DL1+ NK cells was observed in patients with an MPR [32]. The baseline NKG2D+ NK cell levels were associated with nodal involvement, whereas NKG2D+ expressing NK-like cells and T cells were associated with the absence of nodal involvement. The NEOSTAR trial found an increase in CD4^+^ (non-T-reg) CD103^+^ and CD8^+^CD103^+^ T_RM_ and effector memory T cells in patients treated with the ipilimumab-nivolumab combination compared to nivolumab monotherapy, but this independent of the MPR [27].

The T cell receptor (TCR) repertoire and frequency increased after double immunotherapy; however, there was no correlation with pathological response in NEOSTAR [27]. In one patient in CHECKMATE-159, T cell clones specific for tumour neoantigens were expanded in their lymph nodes and peripheral blood after neoadjuvant nivolumab, supporting the hypothesis that neoadjuvant immunotherapy can generate a systemic immune response capable of eradicating micrometastases [22].

As for particular genomic alterations, in the single-arm phase II NCT02716038 trial (chemoimmunotherapy with atezolizumab), adenocarcinoma subtypes with *STK11* mutations demonstrated an inferior outcome after treatment [34]. Moreover, in CHECKMATE-816, circulating tumor DNA (ctDNA) clearance was increased in the chemoimmunotherapy group (56% vs. 35%), and the EFS rate was improved in these patients, regardless of the treatment regimen [30]. A ctDNA analysis from NADIM reported improved survival in patients with low baseline ctDNA and improved survival in those with ctDA clearance after neoadjuvant treatment [33].

Further phase III trials evaluating perioperative neoadjuvant chemoimmunotherapy have shown positive DFS results, including KEYNOTE-671 (pembrolizumab) [36], AEGEAN (durvalumab) [37] and Neotorch (toripalimab) [38]. Biomarker analyses from these trials may further our understanding of the neoadjuvant component of treatments. A summary of the trials discussed and exploratory biomarkers is provided in Table 1.

## 3. Melanoma

Patients with stage III cutaneous melanoma and nodal involvement have 5-year survival rates that range from 93% in stage IIIA to 32% in stage IIID disease, according to data from the American Joint Committee on Cancer (AJCC) [39]. Current management consists of complete resection followed by adjuvant pembrolizumab or nivolumab in *BRAF*wt patients [40]. Neoadjuvant immunotherapy was first tested in a phase I trial by Tarhini et al., with ipilimumab in combination with high-dose interferon alpha, which reported a pCR rate of 32% [41].

A phase 1b trial by Huang et al. (single dose of neoadjuvant pembrolizumab in stage IIIB/C or IV melanoma) reported a pCR rate of 19%. The tumour response was observed in patients with a diffuse tumour-infiltrating lymphocyte (TIL) pattern and was associated with baseline Ki67 expression in non-naïve CD8 T cells. It is also hypothesized that a single dose of anti-PD-1agent leads to the rapid invigoration of exhausted pre-primed CD8 T cells within a week [42]. In a phase II trial by Amaria et al. that tested nivolumab versus a combination of ipilimumab and nivolumab (NCT02519322), higher baseline and on-treatment lymphoid markers and CD8 T cell infiltration were associated with response in both arms of patients [43]. Similarly, low tumour CD8 T cell infiltration was negatively correlated with response in OPACIN (neoadjuvant ipilimumab plus nivolumab) [44].

The phase 1b OPACIN trial comparing perioperative versus adjuvant ipilimumab and nivolumab found an increase in both pre-existing and new T cell clones after neoadjuvant treatment. Patients who relapsed after the neoadjuvant regimen had less expansion of T cell clones [44]. Compared to nivolumab monotherapy, which was associated with an expansion of pre-existing T cell clones in responders, the response to the combination of nivolumab and ipilimumab seemed to be less dependent on the baseline T cell repertoire in the phase II trial by Amaria et al. [43].

An 18-gene IFN-gamma T cell inflamed score was correlated with response in the phase 1b trial by Huang et al. Similar findings were reported in OPACIN and OPACIN-NEO (the subsequent phase II trial evaluating three dosing regimens of the combination of ipilimumab and nivolumab), in which a high IFN-gamma score was also correlated to an increased EFS rate [42,44,45]. Another 10-gene IFN-gamma expression signature was performed on the OPACIN-NEO cohort and showed a positive correlation with the EFS, with potential for daily clinical practice with a short 2-day turn-around time [46]. Low tumour T cell infiltration, lower MHC expression and low PD-L1 expression were identified as negative markers in OPACIN [44].

Tumour cell PD-L1 expression is not associated with response in OPACIN-NEO [45]. Higher tumour cell PD-L1 expression was associated with tumour response after neoadjuvant nivolumab or the combination of ipilimumab and nivolumab [43].

OPACIN-NEO reported a positive correlation between the TMB and EFS (the 2-year EFS was 93.3% in TMB > median versus 58.8% in TMB < median) [47]. TMB is not associated with response in the phase 1b trial by Huang et al. [42,44]. The response was improved regardless of the TMB in the phase II trial by Amaria et al., although higher levels of TMB were described in responders [43].

Data from OPACIN-NEO suggest that the combination of the IFN gamma score and the TMB can predict EFS and response compared to either biomarker alone (AUC 0.83 for the combination of IFN gamma and TMB versus 0.67 for the IFN gamma score and 0.76 for the TMB). Patients with a low TMB/low IFN-gamma score had a particularly poor 2-year EFS of 49.5%.

LDH levels and BRAF status were not associated with recurrence, according to Huang et al. and in OPACIN NEO [42,47].

Another potential biomarker identified in OPACIN NEO is vascular endothelial growth factor receptor (VEGFR-2), which is increased in the plasma of non-responders [47] and can directly induce immunosuppression through T-reg stimulation [48].

A biomarker analysis is expected from the phase II SWOGS1801 trial comparing perioperative versus adjuvant pembrolizumab in locally advanced melanoma, which shows a beneficial effect on the event-free survival in the perioperative arm. Ongoing trials include DOMINI, a biomarker-driven trial which stratifies patients according to their IFN-gamma score and investigates the combination of ipilimumab, nivolumab and a class I histone deacetylase inhibitor, domatinostat (DOM). It is also hypothesized that the IFN gamma score can be increased with a histone deacetylase inhibitor [49]

A third arm of the phase II trial by Amaria et al. (NCT02519322) tested the neoadjuvant relatlimab (anti-lymphocyte-activation gene 3 (LAG-3), a marker for T cell exhaustion) in combination with nivolumab. The biomarker analysis was limited due to small patient numbers, but the baseline PD-L1 and LAG3 did not correlate to response [50].

Intralesional immunotherapy has also been proposed as a therapeutic option. Daromun is a combination immunocytokine consisting of monoclonal L19 antibody fused to TNF-alpha and IL-2 and has been investigated in a phase II trial in non-resectable stage IIIB-IVA melanoma with reported systemic antitumoural effects [51]. Two phase III trials are ongoing in the US and Europe (NEODREAM NCT03567889 and PIVOTAL NCT02938299), investigating neoadjuvant intratumoural Daromun versus upfront surgery.

The NADINA trial (NCT04949113) is a phase III trial comparing neoadjuvant ipilimumab–nivolumab versus adjuvant nivolumab. The experimental arm will receive adjuvant nivolumab in cases of a partial or non-response. The ongoing blPRADO trial is an extension cohort of the phase II OPACIN-NEO which investigates treatment de-escalation (therapeutic lymph dissection and adjuvant treatment) after neoadjuvant combination immunotherapy (ipilimumab 1 mg/kg and nivolumab 3 mg/kg regimen) in case of pCR or MPR. The biomarker analyses will be reported in a later study [52]. A summary of the trials discussed and exploratory biomarkers is provided in Table 2.

## 4. Bladder Cancer

Neoadjuvant cisplatin-based chemotherapy followed by radical cystectomy with lymph node dissection is the standard of care for localized muscle-invasive bladder cancer (MIBC) [53,54]. However, relapse rates are high, and many patients are considered cisplatin ineligible. Cisplatin-based treatments achieve complete pathological response rates ranging from 38% with dose-dense methotrexate vinblastine doxorubicin cisplatin (ddMMVAC) [55] to 21% with cisplatin–gemcitabine [56]. Neoadjuvant immunotherapy in both cisplatin-eligible and non-eligible patients is being tested.

Many trials have reported that PD-L1 expression does not correlate with response [57,58,59,60]. The phase II DUTRENEO trial prospectively classified patients as having “hot” tumours or “cold” tumours depending on an 18-gene IFN-gamma immune signature (tumour immune score, TIS) and tested durvalumab plus tremelimumab versus chemotherapy in the “hot” tumours. Low-PD-L1 patients in the durvalumab plus tremelimumab arm had a notably lower pCR than high-PD-L1 patients (14.3% versus 57%), and there was no difference in pCR in the patients with “hot” tumours treated by chemotherapy. The association of a “hot” tumour TIS score and low PD-L1 expression could be a negative composite biomarker for neoadjuvant immunotherapy [61]. The phase II PURE-01 trial testing single-agent perioperative pembrolizumab reported a positive gain in the EFS in PD-L1-positive patients [62,63]. The 36-month EFS was reported to be 89.8% in the high-PD-L1 patients, compared to 59.7% and 76.7% in the low- and intermediate-PD-L1 combined positive score (CPS), respectively [64]. A baseline ctDNA analysis in the ABACUS trial also showed that PD-L1-negative patients had a particularly poor outcome with neoadjuvant atezolizumab monotherapy [65]. The NABUCCO trial also reported a numerically better complete pathologic response rate to double immunotherapy (ipilimumab–nivolumab) in PD-L1-positive patients, although it was non-significant (73% vs. 33%) [66].

Positive correlations with the TMB have been reported in HCRN GU16-257 (perioperative nivolumab combined with cisplatin gemcitabine), NABUCCO and NCT02989584 (perioperative cisplatin gemcitabine atezolizumab) [66,67,68]. Increasing TMB levels were also positively correlated with the overall EFS in PURE-01; however, the association was weaker than with the CPS [64]. No correlation was seen in NCT02812420 (perioperative durvalumab + tremelimumab), ABACUS or BLASST1 (neoadjuvant nivolumab plus chemotherapy) [58,59,60].

The phase II ABACUS trial testing neoadjuvant atezolizumab reported an increased pCR (40% vs. 20%) in patients with high versus no CD8 intraepithelial infiltration. This finding was also correlated with the RFS. Interestingly, pre- and post-treatment analyses showed that, unlike responding tumours, relapsing tumours did not demonstrate an increase in CD8 infiltration [59,65]. Analyses of tissue samples also demonstrated that the quality of T cell infiltration, as assessed by Granzyme B staining, was correlated with response. In inflamed tumours, defined as having high lymphocytic infiltration, dual granzyme B and CD8 staining was more frequent in responding (87%) versus relapsing tumours (30%).

In ABACUS, an 8-gene T cell signature (tGE8) score at baseline was increased in responding patients. Patients with the tGE8 signature and dual CD8–granzyme B expression had a 40% response rate [59]. The same signature was positively correlated with tumour response in NCT02989584 (cisplatin gemcitabine atezolizumab) [67]. However, tGE8 was not associated with response in NABUCCO and NCT02812420 (durvalumab plus tremelimumab) [60,66].

A molecular subtype analysis reported in BLASST-1 showed increased downstaging rates in basal-type tumours compared to those of luminal-type tumours [58]. The PURE-01 trial also reported an increased EFS rate in basal/squamous and claudin-low molecular subtypes [64].

The presence of a pretreatment tertiary lymphoid structure (TLS) has been associated with response in patients treated with durvalumab and tremelimumab (DUTRENEO) [61]. A 4-gene TLS signature (*POU2AF1*, *LAMP3*, *CD79A* and *MS4A1*) was described as significantly more highly expressed in patients that respond to neoadjuvant immunotherapy [60]. Although baseline TLS was not associated with response in the NABUCCO study, both studies reported an increase in TLSs in responding patients [61,66].

Stromal TGF-beta and fibroblast activation protein (FAP) were correlated with tumour relapse and non-response in NABUCCO and ABACUS [59,66].

Ongoing phase III trials include KEYNOTE-905 EV-303 (NCT03924895), which compares perioperative pembrolizumab in combination with enfortumab vedotin (EV), perioperative pembrolizumab monotherapy and surgery alone in cisplatin-ineligible patients. Similarly, the VOLGA trial will evaluate perioperative durvalumab in combination with tremelimumab and EV versus durvalumab in combination with EV. In cisplatin-eligible patients, KEYNOTE-866 is testing perioperative pembrolizumab versus a placebo in combination with cisplatin–gemcitabine neoadjuvant chemotherapy (NCT03924856). The NIAGARA trial compares neoadjuvant cisplatin–gemcitabine with perioperative durvalumab or a placebo. The ENERGIZE trial will test perioperative linrodostate mesylate (an IDO1 inhibitor) in combination with nivolumab and cisplatin–gemcitabine [69]. IDO1, through kynurenine production, upregulates T-reg. The dual inhibition of IDO1 and PD-L1 has the potential for synergistic antitumour activity. The phase II AURA trial will compare neoadjuvant chemotherapy (cisplatin combined with gemcitabine, ddMVAC regimen or paclitaxel–gemcitabine in cisplatin-ineligible patients) with or without avelumab. This trial will include patients treated with ddMVAC as per the VESPER trial, which reported an improved local control rate compared to that of cisplatin–gemcitabine [55,70]. A summary of the trials discussed and exploratory biomarkers is provided in Table 3.

## 5. Triple-Negative Breast Cancer

Triple-negative breast cancer is the subtype associated with the lowest survival rates. Early stage II and III patients face a high relapse rate despite taxane- and anthracyline-based neoadjuvant chemotherapy [71].

The phase III KEYNOTE 522 trial investigated neoadjuvant chemoimmunotherapy (carboplatin plus paclitaxel followed by adriamycin plus cyclophosphamide, combined with pembrolizumab) and adjuvant pembrolizumab in early triple-negative breast cancer and reported a 64.8% pCR rate versus 51.2% in the group treated with chemotherapy alone [72]. These results led to the FDA approval of pembrolizumab in the neoadjuvant setting with chemotherapy. Another phase III trial, IMPASSION031, reported a 58% pCR rate in patients treated with chemoimmunotherapy (nab-paclitaxel followed by doxorubicin plus cyclophosphamide, combined with atezolizumab) versus 41% in patients treated with chemotherapy alone [73]. On the other hand, the phase III NeoTRIPaPDL1 study did not report a significant increase in pathological response after adding atezolizumab to neoadjuvant chemotherapy (nab-paclitaxel plus carboplatin) [74]. Biomarkers for response to neoadjuvant immunotherapy in early TNBC have not yet been identified.

PD-L1 was not correlated to response in KEYNOTE-522 [72]. In IMPASSION031, response to atezolizumab was seen regardless of PD-L1 levels, although the response to atezolizumab was seemingly increased in PD-L1-positive patients [73]. A positive correlation between response and PD-L1 levels was reported in GeparNuevo in the overall group (phase II trial of neoadjuvant durvalumab combined with nab-paclitaxel followed by epirubicin plus cyclophosphamide); however, this was not specific to durvalumab (the pCR rate in PD-L1-positive tumours was 58% versus 44.4% in the PD-L1-negative group (*p* = 0.445)) [75]. In NeoTRIP, the pCR was significantly higher in PD-L1-positive patients in the durvalumab arm compared to that of the placebo arm [74].

Data concerning the TMB as a potential biomarker for neoadjuvant immunotherapy in TNBC are scarce. A subgroup analysis in GeparNuevo showed that a high baseline TMB was associated with the pCR in the overall cohort and was not specific to the durvalumab group [76].

Stromal TILs have been demonstrated to predict a better response to neoadjuvant chemotherapy [77]. GeparNuevo supports these findings, with an increase in pathological response in patients with higher baseline sTILS in the overall cohort and across both treatment arms; this was not specific to immunotherapy [75]. Post-treatment analyses in GeparNuevo suggest that patients with residual disease and high post-treatment TILs had better outcomes [78]. Unsurprisingly, single-arm phase II NeoPACT (neoadjuvant pembrolizumab combined with carboplatin plus docetaxel, without anthracycline) also reported a significant increase in the pCR in the low- versus high-baseline sTILS [79]. Being a single-arm study, it is unknown whether the positive outcome is associated with the addition of immunotherapy.

Exploratory analyses from GeparNuevo have suggested that seven genes implicated in IFN signaling and antigen presentation are correlated to response in the durvalumab *HLA-A*, *HLA-B*, *TAP1*, *GBP1*, *CXCL10*, *STAT1* and *CD38*. The predefined GeparSixo immune activation and IFN signatures were associated to better responses in the overall cohort and were not identified as specific markers for response to neoadjuvant durvalumab [80]. DetermaIO is a 27-gene expression profile that classifies the TME as a “hot” or “cold” tumour. In eTNBC, it was tested in NeoPACT and in a phase I/II trial by Foldi et al. (neoadjuvant durvalumab plus nab-paclitaxel and doxorubicin/cyclophosphamide) and has been correlated with increased pCR [79,81]. Data from NeoPACT show an 81% pCR in DetermaIO-positive patients versus 43% in DetermaIO-negative patients [79]. Whether this increase in pCR is specific to the addition of immunotherapy to neoadjuvant chemotherapy and whether this assay can be used as a biomarker have yet to be analysed in larger randomized trials. I-SPY-2 is a multiplatform phase II trial comparing multiple investigational arms to a standard chemotherapy paclitaxel–anthracycline control arm in early breast cancer. The addition of pembrolizumab to chemotherapy in the triple-negative group was associated with pCR rates of 60% versus 22% in the control arm [82]. Among 27 different mechanism-of-action-based gene signatures, dendritic cells and STAT1/chemokine signatures were identified as being the most predictive of pCR in the triple-negative group receiving pembrolizumab [83].

Another potential biomarker could be major histocompatibility complex (MHC)-II expression, which has been retrospectively investigated in the I-SPY-2 pembrolizumab arm and in NeoPACT. The MHC-II expression was specifically higher among responders in the pembrolizumab group than in the control group, with an AUC of 0.73 (*p* = 0.001) [84].

Nodal status had no statistically significant impact on pCR in KEYNOTE 522 and IMPOWER031 or on EFS in KEYNOTE-522 [72,73,85].

GeparDouze (NCT03281954) is an ongoing phase III trial evaluating neoadjuvant chemoimmunotherapy with carboplatin plus paclitaxel followed by an anthracycline and cyclophosphamide, combined with atezolizumab. Neoadjuvant immunotherapy is also being tested in ERBB2-negative early breast cancer in combination with olaparib (I-SPY), in ERBB-2-positive breast cancer (neoPATH, IMPASSION 051) and hormone-receptor-positive breast cancer (GIADA trial). A summary of the trials discussed and exploratory biomarkers is provided in Table 4.

## 6. Gastric Cancer

Operable gastric cancer is treated by surgery and perioperative chemotherapy. However, its perioperative morbidity is high, as is the relapse risk. MMR deficiency has been identified as a potential negative biomarker for patients receiving perioperative fluoropyrimidine–platinum-based chemotherapy. OS is worse in dMMR receiving chemotherapy (at 9.6 months) when compared to those treated with surgery alone (11.5 months, median OS not reached), whereas pMMR patients treated with perioperative chemotherapy have better outcomes (with an OS of 19.5 months) [86]. In light of the positive response to pembrolizumab in dMMR patients in the metastatic setting, immunotherapy has been tested in this population in the neoadjuvant setting [87].

The GERCOR NEONIPIGA perioperative single-arm phase II trial investigated combination perioperative ipilimumab–nivolumab in MSI-H/dMMR patients with resectable gastric or gastroesophageal junction adenocarcinoma and showed a 100% R0 resection rate and a 58.6% pCR rate. Three of the twenty-nine patients did not require surgery. Albeit a small trial, CPS was not identified as a potential predictor of tumour response [88]. The phase IIb DANTE trial investigates chemoimmunotherapy (atezolizumab plus FLOT regimen) versus chemotherapy alone. Its preliminary results suggest a positive correlation of CPS with response. In MSI-H patients, pathological regression seemed to improve across both treatment arms and was reported to be 50% in the chemoimmunotherapy group. NEO-PLANET is a phase II trial testing neoadjuvant camrelizumab plus chemoradiotherapy in locally advanced gastric/GEJ adenocarcinoma. TMB was identified as a potential biomarker, whereas the response rate was not altered by PD-L1 expression [89].

Ongoing III trials testing perioperative chemoimmunotherapy versus perioperative chemotherapy include KEYNOTE 585 (with pembrolizumab) [90], which has an improved pCR without improving the EFS rate according to a recent press release, and MATTERHORN (with durvalumab) [91]. A summary of the trials discussed and exploratory biomarkers is provided in Table 5.

## 7. Colorectal Cancer

The standard of care in colorectal cancer is surgery followed by adjuvant chemotherapy in stage III and high-risk stage II patients [92]. Neoadjuvant chemotherapy in locally advanced (cT3 and N+) colorectal cancer was evaluated in a phase III trial and reported low response rates (a pCR of 3.8% and a near-complete tumour regression of 4.6%) after 6 weeks of preoperative FOLFOX [93]. In locally advanced rectal cancer (LARC), chemoradiotherapy is followed by total mesorectal surgery. Its perioperative morbidity is high and includes bowel, urinary and sexual dysfunction.

MMR deficiency is the most prominent biomarker candidate in colon and rectal cancers. NICHE-1 is a phase II trial that compared neoadjuvant ipilimumab–nivolumab in pMMR and dMMR patients with resectable colorectal cancer. In the dMMR patients, the addition of celecoxib to ipilimumab and nivolumab was also tested. This trial reported a 60% pCR rate and a 100% pathological response rate in dMMR after neoadjuvant immunotherapy versus a 27% pathological response rate in pMMR tumours. The addition of celecoxib did not improve the response rate [94]. This trial identified microsatellite instability as a positive marker for response in the neoadjuvant setting. The subsequent phase II single-arm NICHE2 trial, which investigated ipilimumab plus nivolumab in dMMR patients with cT3 or N+ colorectal cancer, reported a 67% pCR and a 99% pathological response rate [95]. In rectal cancer, Cercek et al. reported a 100% (12 patients) clinical complete response rate in dMMR patients with stage II/III rectal adenocarcinoma after 6 months of monotherapy with dostarlimab (anti-PD1 therapy). Standard therapy (neoadjuvant chemoradiotherapy followed by surgery) was omitted in all patients, thus increasing their quality of life [96]. Nivolumab was tested after chemoradiotherapy for locally advanced rectal cancer in the EPOC 1504 VOLTAGE phase I/II study in MSS and MSI-H patients and showed pCR rates of 30% and 60%, respectively [97]. An ongoing phase II trial among dMMR/MSI-H patients with LARC is examining neoadjuvant nivolumab–ipilimumab combined with hypofractionated radiotherapy [98].

Further biomarker analyses have been reported in pMMR populations with both colon and rectal adenocarcinoma receiving neoadjuvant immunotherapy. TMB was not associated with response in NICHE-1 in pMMR patients [94]. A positive correlation has, however, been reported in VOLTAGE and in the phase II trial by Lin et al. (preoperative radiotherapy followed by chemotherapy and camrelizumab in LARC, NCT04231552) [97,99]. In NICHE1, the presence of baseline CD8+ PD-1+ T cells predicted response in pMMR tumours. However, TCR clonality, CD3+, CD8+ and FOXP3+ TCI, IFN-γ score, TLS presence and CXCL13 expression were not associated with response [94]. In VOLTAGE, PCR was correlated with a high ratio of CD8 to effector regulatory cells in TILS. Patients with a consensus molecular subtype (CMS) 1 (i.e.,: “immune”) in VOLTAGE achieved higher PCR rates comparable to those of other CMSs [97].

The results of cohort D with nivolumab and ipilimumab from the VOLTAGE trial and biomarker analyses from the AVANA trial (phase II, avelumab plus chemoradiotherapy in LARC) will be available in the future. A summary of the trials discussed and exploratory biomarkers is provided in Table 6.

## 8. Discussion

While some factors appear to be predictive of response to neoadjuvant immunotherapy (Figure 1), no single predictive biomarker for neoadjuvant immunotherapy has been identified. In contrast to the metastatic setting, PD-L1 expression has no clear correlation with response to immunotherapy in the neoadjuvant setting. MMR deficiency in colorectal cancers stands out as the most impactful biomarker. Despite the impressive 100% complete clinical response rate reported by Cercek et al., response rates were not as high in a similar trial, albeit after prior neoadjuvant chemoradiotherapy (VOLTAGE). Furthermore, the prevalence of MMR deficiency is low, representing 5% of patients with rectal adenocarcinoma [100]. In line with the biological rationale that neoadjuvant immunotherapy relies on an intact IFN-gamma pathway, according to Liu et al., promising biomarkers include interferon-gamma pathway gene signatures. There is, however, no standard assessment tool for the interferon-gamma pathway. The role of B cells in neoadjuvant immunotherapy through the formation of TLS is not yet fully understood and could lead to potential biomarkers. It is possible that biomarkers in the metastatic and the neoadjuvant settings are different, owing to different immune landscapes.

Neoadjuvant immunotherapy trials still face multiple challenges. The currently used endpoints for neoadjuvant immunotherapy are pCR and MPR, which have been approved as surrogate endpoints for OS and EFS in NSCLC and early TNBC following neoadjuvant chemotherapy regimens [101,102]. However, they are yet to be formally validated for neoadjuvant immunotherapy. This is an important issue, as most ongoing phase III trials with immunotherapy use perioperative immunotherapy (Table 7) and will highlight the efficacy of the neoadjuvant portion of the therapy based on pCR. The EFS and OS rates could be impacted by the adjuvant therapy as well. Furthermore, these surrogate endpoints have been assessed the same way as for neoadjuvant chemotherapy. A pathologic assessment of residual viable tumours (RVTs) could be evaluated according to the newly proposed immune-related pathologic response criteria (irPRC) by Cottrell et al. These criteria consider the “regression bed” and include immune-mediated regression features [103]. A common finding in immunotherapy trials is that radiologic response evaluations underestimate pathological response [45,66,94]. Radiomics have the potential improve radiological response prediction and have shown promising results in early lung cancer [104,105]. At this time, pathologic evaluations are still necessary to guide management.

Identifying a single biomarker in the neoadjuvant setting to guide decision making seems unlikely. Future research should therefore integrate multimodal variables, using methods such as artificial intelligence to develop comprehensive classifiers. In early triple-negative breast cancer, using machine learning to process pretreatment data (histology, clinical information and radiology) has the potential to predict whether patients are likely to have a pCR after neoadjuvant chemotherapy and furthermore to inform PFS [106]. We await such analyses for neoadjuvant immunotherapy.

Neoadjuvant immunotherapy treatment regimens have yet to be optimized, regarding preoperative timing, treatment schedules, doses, combinations with other ICPIs or chemotherapy, the choice of the chemotherapy backbone and the addition of adjuvant treatment. In particular, the chemotherapy backbone has the potential to affect immunotherapy efficacy through tumour microenvironment changes. The phase II TONIC trial tested chemotherapy induction before nivolumab treatment in metastatic TNBC and showed a higher response rate with chemotherapy induction, linked to the upregulation of cytotoxic pathways and PD-L1. The response rate was higher with doxorubicin than with cyclophosphamide [107]. As long as the optimal treatment regimen is not identified, biomarker analyses across trials will be inconsistent. In addition, biomarker analyses are not standardized regarding assays and cutoffs. To determine whether a biomarker is predictive, trials should include two arms (with and without treatment) and test for interactions between the biomarker and the treatment group to distinguish between the prognostic and predictive value of the biomarkers. It is difficult to draw conclusions from single-arm studies. Ongoing phase III trials should help provide a better understanding of potential biomarkers.

Neoadjuvant treatment provides a unique opportunity to analyses tumour changes during treatment, giving us the possibility to explore not only baseline biomarkers, but also post-treatment biomarkers. For example, postoperative management could be guided by residual tumour burden, the quality of immune cell infiltration or ctDNA. In NADIM, negative post-treatment ctDNA correlated with increased PFS and OS [33]. Most current trials include an adjuvant immunotherapy phase regardless of tumour response. Therefore, there is potential to evaluate treatment de-escalation after neoadjuvant treatment, both in terms of the necessity for surgery and adjuvant therapy.

Finally, neoadjuvant immunotherapy is a window during which patients are at risk of becoming ineligible for surgery due to disease progression or treatment toxicity. For example, the nivolumab monotherapy arm in the phase II trial by Amaria et al. stopped early due to high progression rates [43]. Treatment discontinuation due to toxicity was reported to be 23% in IMPOWER-031 and KEYNOTE-522 and 10% in CHECKMATE-816 [22,72,74]. Biomarkers in this setting are therefore also necessary to identify patients who will not benefit from neoadjuvant immunotherapy and who should either receive neoadjuvant chemotherapy if applicable or benefit from surgery upfront.

## 9. Conclusions

Neoadjuvant immunotherapy trials have shown promising results, but challenges remain in identifying both patients who will benefit from such approaches and those who will not and risk becoming inoperable. Provided that biomarker analyses are mostly exploratory so far, it would seem that the conventional biomarkers from the metastatic setting are less robust in the neoadjuvant setting. Owing to a different immune landscape, it is possible that other biomarkers are specific in the neoadjuvant setting. Such biomarkers will require validation through prospective biomarker-stratified trials. Future directions will include multi-omic approaches that integrate molecular, clinical and radiological data to improve outcome prediction.

## Figures and Tables

**Figure 1 ijms-24-11849-f001:**
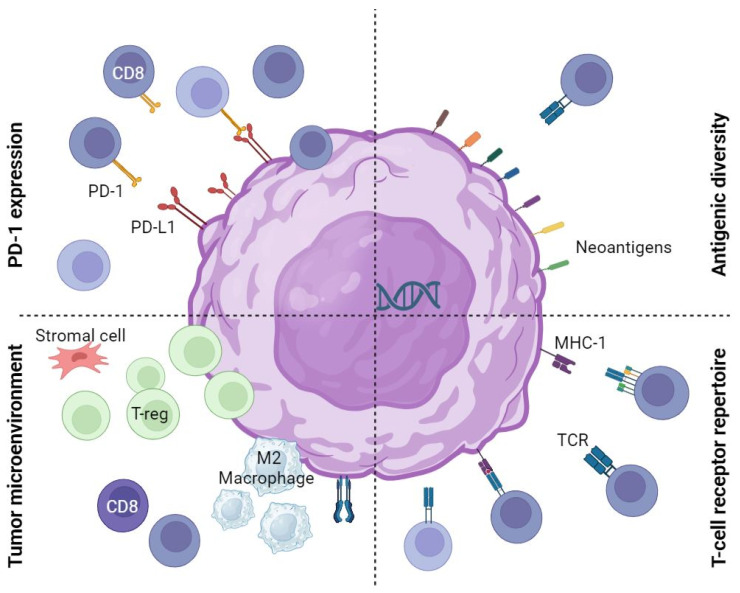
Potential predictors of response to neoadjuvant immunotherapy. PD-1 or PD-L1 expression may influence sensitivity to immunotherapy. The tumour microenvironment, including CD8 T-infiltrating lymphocytes (sensitivity) and regulatory T cells (resistance), may influence the effect of immunotherapy. Antigenic diversity, as seen in high tumour mutation burden or microsatellite instability, may influence sensitivity to immunotherapy. The T cell receptor repertoire or diversity may increase the efficacy of immunotherapy. PD: programmed death, T-reg: regulatory T cell, TCR: T cell receptor, MHC: major histocompatibility complex and CD8: cytotoxic T lymphocytes.

**Table 1 ijms-24-11849-t001:** Summary of exploratory biomarker analyses in neoadjuvant immunotherapy for non-small cell lung cancer.

Study Name or Number	Phase	Regimen	Positive Biomarker	Non-Contributing Biomarker	Negative Biomarker
Checkmate 816	III	Chemotherapy ± nivolumab	PD-L1, non-squamous histology, ctDNA clearance	TMB	-
LCMC3	II	Chemotherapy ± atezolizumab	PD-L1, CD8	TMB, squamous histology	NKG2A expression
NEOSTAR	II	Nivolumab ± ipilimumab	PD-L1, ctDNA clearance	TCR repertoire	-
NADIM	II	Chemotherapy ± nivolumab	-	PD-L1, TMB	-
SAKK16/14	II	Chemotherapy + durvalumab	-	PD-L1	-
NCT02716038	II	Chemotherapy + atezolizumab	-	PD-L1, squamous histology	*STK11mut* subtypes

**Table 2 ijms-24-11849-t002:** Summary of exploratory biomarker analyses in neoadjuvant immunotherapy for melanoma.

Study Name or Number	Phase	Regimen	Positive Biomarker	Non-Contributing Biomarker	Negative Biomarker
OPACIN-NEO	II	Nivolumab + ipilimumab	10 gene IFN-gamma score, TMB	PD-L1	VEGFR-2
NCT02519322	II	Nivolumab ± ipilimumab	PD-L1, CD8 infiltration	TMB	-
OPACIN	Ib	Nivolumab + ipilimumab	CD8 infiltration, T cell expansion, 10 gene IFN-gamma score	-	-
NCT02434354	Ib	Pembrolizumab	18 gene IFN-gamma score	TMB	-

**Table 3 ijms-24-11849-t003:** Summary of exploratory biomarker analyses in neoadjuvant immunotherapy for urothelial cancer.

Study Name or Number	Phase	Regimen	Positive Biomarker	Non-Contributing Biomarker	Negative Biomarker
DUTRENEO	II	Chemotherapy + durvalumab + tremelimumab	PD-L1, tumour immune score (TIS), TLS	-	-
PURE-01	II	Pembrolizumab	PD-L1, TMB, IFN gamma expression	-	-
ABACUS	II	Atezolizumab	PD-L1, CD8-granzyme B expression, tGE8 signature	TMB	Stromal TGF-beta, FAP
HCRN GU16-257	II	Chemotherapy + nivolumab	TMB	-	-
NCT02989584	II	Chemotherapy + atezolizumab	TMB, tGE8 signature	PD-L1	-
BLASST1	II	Chemotherapy + nivolumab	TMB	PD-L1	-
NABUCCO	Ib	Ipilimumab + nivolumab	PD-L1, TMB	tGE8 signature, sTILs, TLS	Stromal TGF-beta, FAP
NCT02812420	I	Durvalumab + tremelimumab	TLS	PD-L1, TMBtGE8 signature	Stromal TGF-beta, FAP

**Table 4 ijms-24-11849-t004:** Summary of exploratory biomarker analyses in neoadjuvant immunotherapy for triple-negative breast cancer.

Study Name or Number	Phase	Regimen	Positive Biomarker	Non-Contributing Biomarker
KEYNOTE-522	III	Chemotherapy ± pembrolizumab	-	PD-L1
IMPASSION031	III	Chemotherapy ± atezolizumab	-	PD-L1
NeoTRIP	III	Chemotherapy ± atezolizumab	PD-L1, sTILs	-
GeparNUEVO	II	Chemotherapy ± durvalumab	PD-L1, sTILs	TMB
NeoPACT	II	Chemotherapy + pembrolizumab	sTILs, DetermaIO profile	-
I-SPY-2	II	Chemotherapy + pembrolizumab	Dendritic cell signature, STAT1/chemokine signature, MHCII expression	-

**Table 5 ijms-24-11849-t005:** Summary of exploratory biomarker analyses in neoadjuvant immunotherapy for gastric cancer.

Study Name or Number	Phase	Regimen	Positive Biomarker	Non-Contributing Biomarker
GERCOR NEONIPIGA	II	Ipilimumab + nivolumab	MSI	PD-L1 CPS
DANTE	IIb	Chemotherapy ± atezolizumab	PD-L1 CPS, MSI	-
NEO-PLANET	II	Chemoradiotherapy + camrelizumab	TMB	PD-L1

**Table 6 ijms-24-11849-t006:** Summary of exploratory biomarker analyses in neoadjuvant immunotherapy for colorectal cancer.

Study Name or Number	Phase	Arms	Positive Biomarker	Non-Contributing Biomarker
NICHE-1	I	Ipilimumab + nivolumab	MSI	TMB, TCR clonality, IFN-gamma score, TLS
NICHE-2	II	Ipilimumab + nivolumab	MSI	-
NCT04165772	II	Dostarlimab	MSI	-
VOLTAGE	II	Chemoradiotherapy then nivolumab	MSI, TMB, CD8 TILs, consensus molecular subtype 1	-
NCT04231552	II	Chemoradiotherapy then camrelizumab	TMB	-

**Table 7 ijms-24-11849-t007:** Selected ongoing phase III trials including neoadjuvant immunotherapy.

Study Name or Number	Cancer Type	Arms	Adjuvant	Primary Endpoint(s)
Checkmate 77T	NSCLC	Neoadjuvant chemotherapy ± nivolumab	Nivolumab	EFS
IMpower030	NSCLC	Neoadjuvant chemotherapy ± atezolizumab	Atezolizumab	MPR
NADINA	Melanoma	Neoadjuvant ipilimumab-nivolumab versus adjuvant nivolumab	Nivolumab (control arm)	EFS
NeoDREAM	Melanoma	Neoadjuvant intralesional Daromun + adjuvant therapy versus adjuvant therapy	Investigator’s choice of adjuvant therapy	RFS
NIAGARA	Bladder	Neoadjuvant chemotherapy ± durvalumab	Durvalumab	pCR and EFS
Keynote 905	Bladder	Neoadjuvant pembrolizumab ± enfortumab vedotin	Pembrolizumab ± enfortumab vedotin	pCR and EFS
GeparDouze	TNBC	Neoadjuvant chemotherapy ± atezolizumab	Atezolizumab	pCR and EFS
MATTERHORN	Gastric-gastro-oesophageal	Perioperative chemotherapy ± durvalumab	Chemotherapy ± durvalumab	EFS
Keynote 585	Gastric-gastro-oesophageal	Perioperative chemotherapy ± pembrolizumab	Chemotherapy ± pembrolizumab	pCR and EFS

## Data Availability

Not applicable.

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
