# Peer review of "Neoadjuvant Immunotherapy: A Promising New Standard of Care"

_ijms, 2023, doi:10.3390/ijms241411849_

Round 1
Reviewer 1 Report
This MS aims to overview potential biomarkers and possible clinical benefits of neo-adjuvant immunotherapy in early triple negative breast cancer, bladder cancer, melanoma, non-small cell lung cancer, colorectal cancer and gastric cancer. Neo-adjuvant treatments make me cautious because patients are at risk of disease progression during this phase. However, it should be recognized that neo-adjuvant immunotherapy can increase the immunizing (vaccinating) effect of the tumor and improve the outcome of the disease. The MS is highly relevant and could be of interest to a diverse group of clinicians and researchers.
My comments and suggestions.
1) Figure 1 has no caption and in this form is not informative.
2) Table 1 is also not informative. Clinical treatment results are not included, only primary endpoint(s). Column 4 “Adjuvant” essentially duplicates column 3 “Arms”. The MS includes various data and needs additional summary conclusions. It might be helpful to include summarizing table in each subsection. The MS also needs a conclusions subsection.
3) Could you explain the theoretical rationale for the simultaneous use of immunosuppressive neo-adjuvant chemotherapy and immunostimulatory neo-adjuvant immunotherapy.
Author Response
Thank you for your insightful comments. Attached is a document in which we address each point you made and how we incorporated changes into the manuscript.

Reviewer 2 Report
The authors have submitted a review article called “Neoadjuvant Immunotherapy: a promising new standard of Care”. They summarized a comprehensive review of biomarkers of various cancers. A few questions shall be answered before further processing.
1. Potential biomarkers and conventional biomarkers (e.g., PD-L1) shall be indicated clearly.
2. Future perspectives should be discussed more in-depth. For example, how novel methods such as image features, machine learning, and multi-omics in exploring biomarkers can influence Neoadjuvant immunotherapy?
3. Challenges should be included as currently Neoadjuvant immunotherapy still faces challenges. Taking NSCLC as an example, determining whether oncogenic-addicted NSCLC can be excluded from single immunotherapy and a number of cycles prior to surgery for maximum benefits, and adopting the appropriate surrogate endpoint of long-term survival in neoadjuvant trials and more are still challenging.
Author Response

(The authors gave the same response as above.)
